# Hepatitis B care cascade among people with HIV/HBV coinfection in the North American AIDS Cohort Collaboration on Research and Design, 2012–2016

Jessica Kim[1], Craig W. Newcomb[2], Dean M. Carbonari[2], Jessie Torgersen[3], Keri N. Althoff[4], Mari M. Kitahata[5], Marina B. Klein[6], Richard D. Moore[4], K. Rajender Reddy[3], Michael J. Silverberg[7], Angel M. Mayor[8], Michael A. Horberg[9], Edward R. Cachay[10], Joseph K. Lim[11,12], M. John Gill[13], Kara Chew[14], Timothy R. Sterling[15], Mark Hull[16], Eric C. Seaberg[4], Gregory D. Kirk[4], Sally B. Coburn[4], Raynell Lang[13], Kathleen A. McGinnis[12], Kelly A. Gebo[4], Sonia Napravnik[17], H. Nina Kim[5], Vincent Lo Re, III[2,3]*, for the North American AIDS Cohort Collaboration on Research and Design of IeDEA[¶]

1 Department of Medicine, Drexel University College of Medicine, Philadelphia, Pennsylvania, United States of America, 2 Department of Biostatistics, Epidemiology and Informatics, University of Pennsylvania Perelman School of Medicine, Philadelphia, Pennsylvania, United States of America, 3 Department of Medicine, University of Pennsylvania Perelman School of Medicine, Philadelphia, Pennsylvania, United States of America, 4 Department of Epidemiology, Bloomberg School of Public Health, Johns Hopkins University, Baltimore, Maryland, United States of America, 5 Department of Medicine, University of Washington, Seattle, Washington, United States of America, 6 Department of Medicine, McGill University Health Centre, Montreal, Quebec, Canada, 7 Division of Research, Kaiser Permanente Northern California, Oakland, California, United States of America, 8 Retrovirus Research Center, Universidad Central del Caribe, Bayamon, Puerto Rico, 9 Mid-Atlantic Permanente Research Institute, Mid-Atlantic Permanente Medical Group, Rockville, Maryland, United States of America, 10 Department of Medicine, University of California San Diego, La Jolla, California, United States of America, 11 Department of Medicine, Yale School of Medicine, New Haven, Connecticut, United States of America, 12 VA Connecticut Healthcare System, West Haven, Connecticut, United States of America, 13 Department of Medicine, University of Calgary, Calgary, Alberta, Canada, 14 Department of Medicine, University of California Los Angeles, Los Angeles, California, United States of America, 15 Department of Medicine, Vanderbilt University Medical Center, Nashville, Tennessee, United States of America, 16 Department of Medicine, University of British Columbia, Vancouver, Canada, 17 Department of Medicine, University of North Carolina at Chapel Hill, Chapel Hill, North Carolina, United States of America

¶ Membership of the North American AIDS Cohort Collaboration on Research and Design of IeDEA is provided in the Acknowledgments.
* vincentl@pennmedicine.upenn.edu

**Data Availability Statement:** The North American AIDS Cohort Collaboration on Research and Design (NA-ACCORD) maintains restrictions on sharing

## Abstract

A care cascade is a critical tool for evaluating delivery of care for chronic infections across sequential stages, starting with diagnosis and ending with viral suppression. However, there have been few data describing the hepatitis B virus (HBV) care cascade among people living with HIV infection who have HBV coinfection. We conducted a cross-sectional study among people living with HIV and HBV coinfection receiving care between January 1, 2012 and December 31, 2016 within 13 United States and Canadian clinical cohorts contributing data to the North American AIDS Cohort Collaboration on Research and Design (NA-ACCORD). We evaluated each of the steps in this cascade, including: 1) laboratory-confirmed HBV infection, 2) tenofovir-based or entecavir-based HBV therapy prescribed, 3) HBV DNA

de-identified data sets. The NA-ACCORD Principals of Collaboration requires submission and approval of a concept sheet that describes the intended research project for which data are being requested. The NA-ACCORD Executive Committee and the Steering Committee (composed of principle investigators from contributing cohorts) must approve the concept sheet and elect to have their data included in the research project. A signed Data User Agreement is required before data can be released. Guidance for how to obtain NA-ACCORD data are outlined on the NA-ACCORD website (https://naaccord.org/collaboration-policies).

**Funding:** This work was supported by a research grant from the National Institute of Allergy and Infectious Diseases [grant number R21 AI124868 (VLR)]. Additional support was provided by National Institutes of Health [grant numbers U01 AI069918 (KNA, RDM), F31 AI124794, F31 DA037788, G12 MD007583, K01 AI093197 (KNA), K01 AI131895, K08 DK132977 (JT), K23 EY013707, K24 AI065298 (TRS), K24 AI118591, K24 DA000432 (RDM), KL2 TR000421, N01 CP01004, N02 CP055504, N02 CP91027, P30 AI027757, P30 AI027763, P30 AI027767, P30 AI036219, P30 AI050409, P30 AI050410, P30 AI094189, P30 AI110527, P30 MH62246, R01 AA016893 (RDM), R01 DA011602 (RDM), R01 DA012568, R01 AG053100 (KNA), R24 AI067039, U01 AA013566, U01 AA020790, U01 AA020793, U01 AI038855, U01 AI038858, U01 AI068634, U01 AI068636, U01 AI069432, U01 AI069434, U01 DA036297 (GDK), R01 DA048063, U01 DA03629, U01 DA036935 (RDM), U10 EY008057, U10 EY008052, U10 EY008067, U01 HL146192, U01 HL146193, U01 HL146194, U01 HL146201, U01 HL146202, U01 HL146203, U01 HL146204, U01 HL146205, U01 HL146208, U01 HL146240, U01 HL146241, U01 HL146242, U01 HL146245, U01 HL146333, U24 AA020794, U54 GM133807, UL1 RR024131, UL1 TR000004, UL1 TR000083, Z01 CP010214 and Z01 CP010176]; Centers for Disease Control and Prevention, USA [contract numbers CDC-200-2006-18797 and CDC-200-2015-63931]; Agency for Healthcare Research and Quality, USA [contract number 90047713]; Health Resources and Services Administration, USA [contract number 90051652]; Canadian Institutes of Health Research, Canada [grant numbers CBR 86906, CBR 94036, HCP 97105 and TGF 96118]; Ontario Ministry of Health and Long Term Care; and the Government of Alberta, Canada. Additional support was provided by the National Institute of Allergy and Infectious Diseases (NIAID), National Cancer Institute (NCI), National Heart, Lung, and Blood Institute (NHLBI), Eunice Kennedy Shriver

measured during treatment, and 4) viral suppression achieved via undetectable HBV DNA. Among 3,953 persons with laboratory-confirmed HBV (median age, 50 years; 6.5% female; 43.8% were Black; 7.1% were Hispanic), 3,592 (90.9%; 95% confidence interval, 90.0–91.8%) were prescribed tenofovir-based antiretroviral therapy or entecavir along with their antiretroviral therapy regimen, 2,281 (57.7%; 95% confidence interval, 56.2–59.2%) had HBV DNA measured while on therapy, and 1,624 (41.1%; 95% confidence interval, 39.5–42.6) achieved an undetectable HBV DNA during HBV treatment. Our study identified significant gaps in measurement of HBV DNA and suppression of HBV viremia among people living with HIV and HBV coinfection in the United States and Canada. Periodic evaluation of the HBV care cascade among persons with HIV/HBV will be critical to monitoring success in completion of each step.

## Introduction

Hepatitis B virus (HBV) infection is common among people with HIV (PWH), with a prevalence ranging from 5–15% [1]. Research has shown that detectable HBV viremia among PWH with HBV coinfection is associated with increased rates of hepatocellular carcinoma (HCC) [2]. HBV suppression with HBV-active antiretroviral therapy (ART) reduces risk of HCC and decompensated cirrhosis [2, 3].

In May 2016, the World Health Assembly adopted the Global Health Sector Strategy on Viral Hepatitis, which called for the elimination of viral hepatitis as a public health threat by 2030 [4]. Elimination was defined as a 90% reduction in incidence and a 65% reduction in the number of related deaths from levels as of 2015 [5]. The strategy addressed all five hepatitis viruses (i.e., hepatitis A, B, C, D, and E), but HBV was of particular focus because of its public health burden.

Monitoring progress towards these global targets across different settings will be crucial to the elimination of HBV infection [6, 7]. Care cascades have emerged as a critical tool for evaluating the delivery of care across sequential stages of management of chronic viral infections, starting with diagnosis of the infection and ending with viral suppression or cure [8–12]. The HIV care cascade (consisting of steps for diagnosis, linkage to care, retention in care, prescription of antiretroviral therapy, and viral suppression) has been an effective tool for improving the health of PWH and for achieving the public health benefits of ART [8, 9]. The hepatitis C virus (HCV) care cascade (consisting of steps for diagnosis, confirmatory HCV RNA testing, prescription of direct-acting antiviral therapy, and viral cure) has been used to assess the delivery of HCV-related care in a variety of settings and has been important for monitoring progress toward HCV elimination goals [10–12]. In contrast, there have been few data describing the HBV care cascade, particularly among PWH with HBV coinfection [13]. This information is crucially important for establishing baseline metrics of HBV care among PWH. These data can also help to identify gaps in HBV management among PWH; enable national, regional, and local agencies to prioritize and target resources to close those gaps; and promote stakeholder involvement and collaboration, all of which support achievement of the 2030 World Health Organization HBV elimination goals [14]. In this study, we describe the HBV care cascade among PWH with HBV coinfection from 2012–2016, which can serve as baseline measures to assess progress toward HBV elimination goals.

National Institute of Child Health & Human Development (NICHD), National Human Genome Research Institute (NHGRI), National Institute for Mental Health (NIMH) and National Institute on Drug Abuse (NIDA), National Institute on Aging (NIA), National Institute of Dental & Craniofacial Research (NIDCR), National Institute of Neurological Disorders And Stroke (NINDS), National Institute of Nursing Research (NINR), National Institute on Alcohol Abuse and Alcoholism (NIAAA), National Institute on Deafness and Other Communication Disorders (NIDCD), and National Institute of Diabetes and Digestive and Kidney Diseases (NIDDK). These data were collected by cancer registries participating in the National Program of Cancer Registries (NPCR) of the Centers for Disease Control and Prevention (CDC). The funders had no role in study design, data collection and analysis, decision to publish, or preparation of the manuscript.

**Competing interests:** J.T. reports grants to her institution from the City of Philadelphia ID/SUD Care Integration Pilot. K.N.A. reports grants to her institution from the National Institute of Health (NIH), royalties from Coursera, and consulting fees from NIH and TrioHealth. M.B.K. reports grants from Canadian Institutes of Health Research, Fonds de recherché Quebec –Sante, NIH, ViiV Health Care, Gilead Sciences, and Abbvie; consulting fees from ViiV Health Care, Gilead Sciences, and Abbvie; leadership role in the CIHR Canadian HIV Trials Network; and receipt of goods/ services from Siga Technologies. K.R.R. reports grants to his institution from Mallinckrodt, Exact Sciences, BMS, Intercept, Merck, Gilead, Grifols, Sequana, HCC-TARGET, NASH-TARGET, and BioVie; royalties from UpToDate; consulting fees from Spark Therapeutics, Mallinckrodt, Genfit, and Novo Nordisk; paid board participation from Novartis; and leadership roles in Gastroenterology and AASLD Task Force for COVID Activities. E.R.C. reports grants to his institution from Gilead Sciences and board participation in THERAtechnologies. J.K.L. reports grants to his institution from Intercept, Gilead, Viking, Pfizer, Eiger, Inventiva, and Novo Nordisk and leadership roles in the American Association for Study of Liver Diseases, American Gastroenterological Association, and American College of Gastroenterology. M.J.G. reports participation on the HIV national advisory boards for Merck, Gilead, and Viiv. K.C. reports grants to her institution from Merck Sharp & Dohme and Amgen; consulting fees from Pardes Biosciences; honoraria payments from International Antiviral Society-USA; and participation in the UCSF Safety Monitoring Committee. M.H. reports grants from Gilead Life

## Methods

### Study design and data source

We conducted a cross-sectional study among PWH with HBV coinfection receiving care between January 1, 2012 and December 31, 2016 within 13 United States (US) and Canadian clinical cohorts contributing data to the North American AIDS Cohort Collaboration on Research and Design (NA-ACCORD). These cohorts include PWH who are engaged in care (≥2 HIV clinic visits within 12 months) [8]. At regular intervals, cohorts collect and securely transfer data (demographic, diagnostic, medication, sociobehavioral, laboratory, vital status information) to the Data Management Core (University of Washington) for harmonization and quality control checks. Data are transferred to the Epidemiology/Biostatistics Core (Johns Hopkins University) for additional quality checks and creation of analytic-ready files. NA-AC-CORD research has been approved by the Institutional Review Boards of each cohort. This nested study was also approved by the University of Pennsylvania Institutional Review Board. We did not have access to information that could identify individual participants during or after data collection, and informed consent was waived given the de-identified nature of these data. The study was conducted between February 2022 and March 2023.

### Study patients

We focused on adult (≥18 years) PWH who were alive with HBV coinfection as of 2012 to assess HBV care when tenofovir-based ART and HBV-specific management guidelines were accessible [15]. PWH in NA-ACCORD were eligible for inclusion if they had: 1) HBV coinfection (defined by ≥1 positive HBV surface antigen, ≥1 positive HBV e antigen, or any detectable HBV DNA) prior to December 31, 2016; 2) ≥365 days of observation in NA-ACCORD after their qualifying HBV laboratory test between January 1, 2012 and December 31, 2016; and 3) ≥1 HIV RNA and CD4+ cell measurement during the 2012–2016 period (to limit inclusion to people in HIV care). All eligible patients were included.

### HBV care cascade steps

Care cascades may be prevalence-based (describing the number of people in each step as a percentage of those estimated to have the condition) or diagnosis-based (describing the number of people in each step as a percentage of those confirmed to have the condition). We evaluated a diagnosis-based cascade because we sought to determine the proportion meeting each step among a denominator of PWH with laboratory-confirmed HBV coinfection. The main steps in this cascade include: 1) laboratory-confirmed HBV infection, 2) tenofovir-based or entecavir-based HBV therapy prescribed, 3) HBV DNA measured during treatment, and 4) viral suppression achieved via undetectable HBV DNA.

For each individual with laboratory-confirmed HBV infection, we determined exposure to a first-line HBV-active antiviral drug (i.e., tenofovir disoproxil fumarate [TDF], tenofovir alafenamide [TAF], or entecavir) from 2012–2016. Since 2010, HIV treatment guidelines have emphasized including tenofovir in the ART regimen of those with chronic HBV and avoidance of lamivudine or emtricitabine monotherapy, which can promote development of HBV drug-resistance mutations [15, 16]. If tenofovir cannot be safely used, entecavir may be administered in addition to fully suppressive ART [15]. ART was defined as receipt of three antiretrovirals from at least two classes [16]. We examined all available quantitative and qualitative HBV DNA results from 2012–2016. We determined age at date of first qualifying HBV test and selected the HIV RNA and CD4+ cell count closest to that date. Sex, race/ethnicity, and HIV transmission risk factors were collected at NA-ACCORD enrollment.

Science for an investigator initiated study and participation in the data safety monitoring board for the M2HepPreP study. G.D.K. reports grants to his institution from NIH. K.A.G. reports grants to her institution from NIH, US Department of Defense, Defense Health Agency, State of Maryland, Octapharma, Mental Wellness Foundation, HealthNetwork Foundation, Bloomberg Philanthropies, and Moriah Fund; royalties from UpToDate; consulting fees from Spark HealthCare, Teach for America, and Aspen Institute; and unpaid advisor participation on a Pfizer scientific advisory board. H.N.K. reports grants to her institution from Gilead Sciences. V.L.R. reports grants to his institution from NIH and a leadership role in the International Society for Pharmacoepidemiology. All other authors reported no conflicts of interest. This does not alter our adherence to PLOS ONE policies on sharing data and materials.

## Statistical analysis

We described the diagnosis-based HBV care cascade as follows:

**Step 1 (HBV Infection)**: Among eligible PWH in NA-ACCORD receiving HIV care between 2012–2016, we determined the number with laboratory-confirmed HBV infection.

**Step 2 (Received Tenofovir or Entecavir)**: We calculated the proportion (with 95% confidence interval [CI]) of PWH with HBV coinfection who, between 2012–2016, received tenofovir-based ART (i.e., TDF or TAF) or entecavir along with an ART regimen.

**Step 3 (HBV DNA Assessed)**: We calculated the proportion (with 95% CI) of PWH with HBV coinfection who had any HBV DNA test (quantitative or qualitative) performed while on HBV treatment (defined in Step 2) between 2012–2016. We determined the median (interquartile range [IQR]) number of HBV DNA measures per patient between 2012–2016 for those who had this test performed. Additionally, to assess whether patients had HIV RNA measured more frequently than HBV DNA, we determined the median number of HIV RNA measures per patient between 2012–2016.

**Step 4 (Undetectable HBV DNA)**: We calculated the proportion (with 95% CI) of PWH with HBV coinfection who had an undetectable quantitative (i.e., HBV DNA <200 international units [IU]/mL) or qualitative (i.e., negative) HBV DNA test during HBV treatment (defined in Step 2) between 2012–2016.

Data were analyzed using SAS Enterprise Guide 8.2 (SAS Institute, Cary, NC).

## Results

Between January 1, 2012 and December 31, 2016, there were 85,546 PWH in care within 13 clinical cohorts in NA-ACCORD. Within this sample, 73,860 (86.3%) PWH ever had HBV laboratory testing, and 5,485 (6.4%) had laboratory-confirmed HBV coinfection. After excluding those with <365 days of observation after the qualifying HBV laboratory test (n = 210) or no available HIV RNA and CD4+ cell measurements during the observation period (n = 1,295), 3,953 people with HIV/HBV coinfection remained in the final sample (Fig 1). These individuals had a median age of 50 (IQR, 43–57) years, 6.5% were female, 43.8% were Black, and 7.1% were Hispanic (Table 1).

Among the 3,953 PWH with HBV coinfection, 3,592 (90.9%; 95% CI, 90.0–91.8%) were prescribed tenofovir-based ART or entecavir along with their ART regimen (Table 1; median time on HBV therapy, 2.7 [IQR, 1.5–4.1] years), 2,281 (57.7%; 95% CI, 56.2–59.2%) had HBV DNA measured while on therapy, and 1,624 (41.1%; 95% CI, 39.5–42.6) achieved an undetectable HBV DNA during HBV treatment (Fig 2). Among those who had HBV DNA measured, 1,624 (71.2%) achieved HBV suppression on therapy. The median number of HBV DNA measures per patient between 2012–2016 for those who had this test performed was 3 (IQR, 2–6). In contrast, the median number of HIV RNA measures per patient between 2012–2016 was 8 (IQR, 4–12).

## Discussion

In this study, we identified gaps in HBV-related management among PWH with HBV coinfection in care in the US and Canada. The largest drop-offs occurred in assessment of HBV DNA and confirmation of HBV suppression on HBV-active therapy. Our study highlights opportunities for improving HBV-related care among PWH with HBV coinfection in a population known to be at risk for more accelerated liver disease progression.

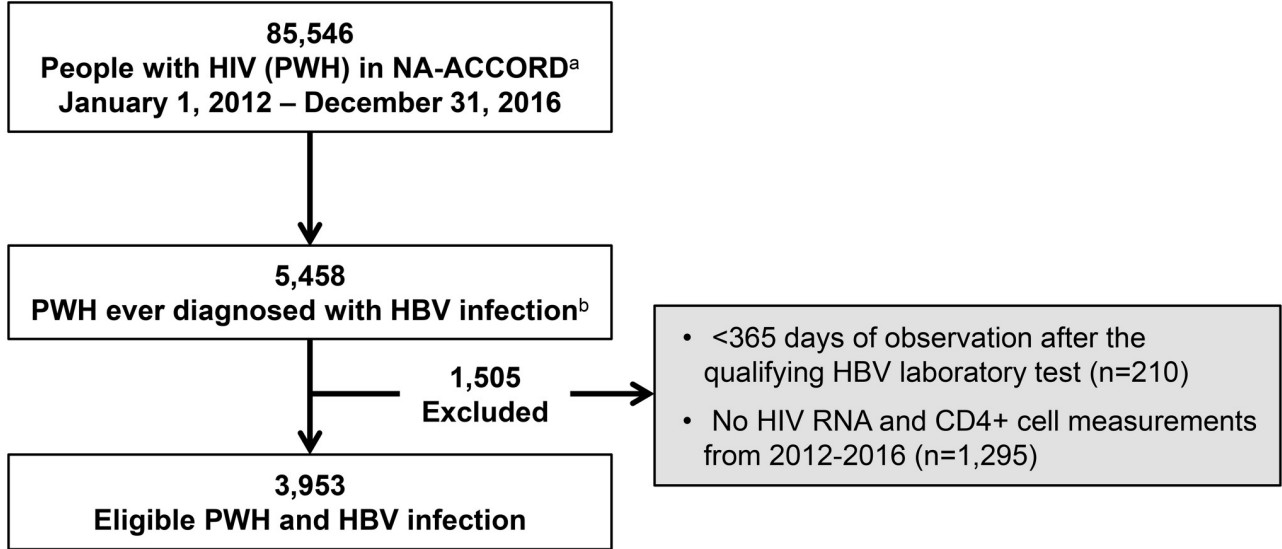

**Fig 1. Selection of people with HIV/hepatitis B virus coinfection within the North American AIDS Cohort Collaboration on Research and Design (2012–2016).** Abbreviations: HBV, hepatitis B virus; HCC, hepatocellular carcinoma; NA-ACCORD, North American AIDS Cohort Collaboration on Research and Design; RNA, ribonucleic acid. [a] Includes 13 contributing clinical cohorts within NA-ACCORD. [b] HBV co-infection determined by positive HBV surface antigen, positive HBV e antigen, or detectable HBV DNA.

We observed that 9.0% of PWH with HBV coinfection did not receive either tenofovir-based ART or entecavir along with their ART regimen. The absence of anti-HBV therapy represents a missed opportunity for prevention of liver complications.

We found that only 57.7% of PWH with HBV coinfection were tested for HBV DNA while on HBV therapy during the observation period. Moreover, the median number of HBV DNA measures was much less than the median number of HIV RNA measures during the 5-year observation period. In a prior study evaluating HBV DNA assessment among 357 people with HIV/HBV coinfection in care at Parkland Health System, Texas from 1999–2003, only 16% had HBV DNA measured [13]. Among a commercially-insured cohort of predominantly individuals with HBV monoinfection, 36% had either HBV DNA or HBV e antigen measured within 12 months after HBV diagnosis [17]. HBV management guidelines suggest that HBV DNA should be assessed every 3–6 months during HBV therapy to confirm HBV DNA suppression [18]. Measuring HBV DNA levels with regular frequency is necessary to assess the response of HBV DNA to antiviral therapy and confirm HBV suppression, a key benchmark associated with improved clinical outcomes [2]. It may also motivate adherence to HBV therapy.

Despite the increased risk of liver complications with elevated HBV DNA levels, only 41.1% of PWH with HBV coinfection on HBV-active ART were confirmed to achieve an undetectable HBV DNA. Among PWH with HBV coinfection who had HBV DNA measured, 71.2% had confirmed HBV suppression. HBV DNA >200 IU/mL is associated with a 2.7-fold higher rate of HCC (hazard ratio = 2.70 [95% CI, 1.23–5.93]) [2]. Moreover, sustained ($\geq$1 year) HBV suppression with HBV-active ART is associated with a 58% reduction in the rate of HCC [2]. Therefore, to ensure the maximal protective benefits from HBV-active ART, providers should be aware of the importance of assessing HBV DNA and confirming HBV suppression. Integrating reminders to measure HBV DNA in electronic medical record systems, creating automated order sets, and provider education may help increase HBV DNA assessments in clinical care [19].

**Table 1. Characteristics of people with HIV/hepatitis B virus coinfection in the North American AIDS Cohort Collaboration on Research and Design (2012–2016).**

| Characteristic | (n = 3,953) |
|---|---|
| **Age (n, %)**[a] | |
| Median (years, IQR) | 49.7 (42.6–57.0) |
| <40 years | 746 (18.9%) |
| 40–49 years | 1,265 (32.0%) |
| ≥50 years | 1,940 (49.1%) |
| **Male sex (n, %)**[b] | 3,696 (93.5%) |
| **Race (n, %)**[b] | |
| White | 1,776 (44.9%) |
| Black or African American | 1,731 (43.8%) |
| Asian/Pacific Islander | 82 (2.1%) |
| Multiracial, Other, Unknown | 364 (9.2%) |
| **Hispanic (n, %)**[b] | 271 (7.1%) |
| **HIV transmission risk factors (n, %)**[b] | |
| Men who have sex with men | 1,466 (37.1%) |
| History of injection drug use | 841 (21.3%) |
| Receipt of blood transfusion, etc. | 7 (0.2%) |
| Heterosexual contact | 379 (9.6%) |
| Other | 70 (1.8%) |
| Unknown | 1,305 (33.0%) |
| **HIV RNA (n, %)**[c] | |
| Median ($\log_{10}$ copies/mL, IQR) | 1.7 (1.3–3.0) |
| ≤75 copies/mL | 2,522 (63.8%) |
| >75 copies/mL | 1,431 (36.2%) |
| **Absolute CD4+ cell count (n, %)**[c] | |
| Median (cells/mm$^3$, IQR) | 430 (245–647) |
| ≥500 cells/mm$^3$ | 1,599 (40.5%) |
| 200–499 cells/mm$^3$ | 1,578 (39.9%) |
| <200 cells/mm$^3$ | 776 (19.6%) |
| **CD4+ cell percentage (n, %)**[c] | |
| Median (%, IQR) | 24.1 (16.0–33.0) |
| ≥28% | 1,571 (39.7%) |
| 14–27.99% | 1,591 (40.2%) |
| <14% | 791 (20.0%) |
| **HBV DNA (median, IQR; $\log_{10}$ IU/mL)**[d] | 1.7 (1.3–2.9) |
| **HBV therapy (n, %)**[e] | |
| Tenofovir[f] + (lamivudine or emtricitabine) | 3,066 (77.6%) |
| Tenofovir[f] alone | 512 (13.0%) |
| Lamivudine or emtricitabine alone | 213 (5.4%) |
| Entecavir | 21 (0.5%) |

(*Continued*)

**Table 1.** (Continued)

| Characteristic | (n = 3,953) |
|---|---|
| Not prescribed HBV therapy | 141 (3.6%) |

Abbreviations: ART = antiretroviral therapy; HBV = hepatitis B virus; HIV = human immunodeficiency virus; IQR = interquartile range; IU = international units; RNA = ribonucleic acid

[a] Age was measured as year of qualifying HBV test—year of birth.

[b] Sex, race/ethnicity, and HIV transmission risk factors were collected at enrollment into the NA-ACCORD.

[c] HIV RNA, CD4+ cell count, and CD4+ cell percentage were selected from dates closest to the first qualifying HBV test (defined by positive HBV surface antigen, positive HBV e antigen, or detectable HBV DNA).

[d] Median HBV DNA calculated using first available HBV DNA measured during 2012–2016 observation period.

[e] Based on prescriptions for HBV antivirals during 2012–2016 observation period. Persons ever prescribed tenofovir + (lamivudine or emtricitabine) during the observation period were classified in this group. Persons never prescribed combination HBV therapy were evaluated for ever use of tenofovir alone or entecavir. Persons not prescribed any of the three aforementioned regimens were then evaluated for use of lamivudine or emtricitabine. The remaining individuals were classified as not having received HBV therapy.

[f] Includes tenofovir disoproxil fumarate or tenofovir alafenamide.

We did not include HCC surveillance as a step in our HBV care cascade because not all people with HBV coinfection are currently recommended to undergo HCC surveillance. According to guidelines by the American Association for the Study of Liver Diseases [20], the European Association for the Study of the Liver and European Organization for Research and Treatment of Cancer [21], and the Asian-Pacific Association for the Study of the Liver [22], HCC surveillance is only recommended for patients with chronic HBV infection and cirrhosis as well as those without cirrhosis who have specific characteristics, such as family history of HCC or certain age thresholds, sex, or race. Thus, since HCC surveillance is not currently recommended for all PWH with HBV coinfection, we did not include it in the cascade.

Our study has several potential limitations. First, we did not assess adherence to HBV therapy. Measures of adherence to antiviral therapies, such as ART or HCV therapy, that are assessed within observational studies (e.g., self-report or pharmacy-based refill measures) can be inaccurate and are typically validated against the gold standard of viral suppression [23–26]. Consequently, as the final step of our HBV care cascade, we assessed the proportion of PWH with HBV coinfection who had an undetectable HBV DNA test during HBV treatment. Future studies should determine the levels of adherence to HBV-active ART or entecavir required to achieve HBV DNA suppression in PWH, which could serve as a target for patients

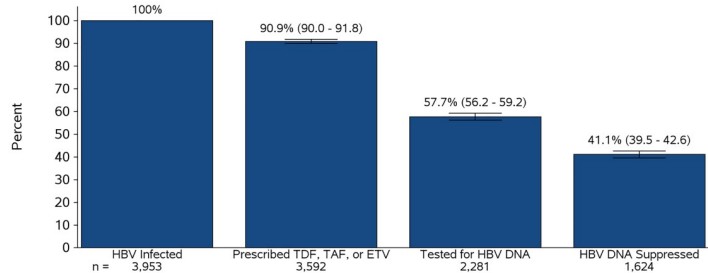

**Fig 2. Cascade of care among HIV/hepatitis B virus-coinfected persons within the North American AIDS Cohort Collaboration on Research and Design (2012–2016).** Abbreviations: ETV, entecavir; HBV, hepatitis B virus; TDF, tenofovir disoproxil fumarate; TAF, tenofovir alafenamide.

to maximize their response to HBV therapy. Second, adefovir or telbivudine may be prescribed for HBV treatment, but these antivirals are not collected by NA-ACCORD. However, they are used infrequently in most settings, particularly among PWH. Third, this study utilized data from 2012–2016 and may not entirely represent current practice. However, HBV management guidelines in HIV have remained largely unchanged during the intervening time, arguing for continued relevance of trends observed in this cascade of care. Finally, NA-ACCORD cohorts included in this analysis represent US and Canadian demographics of PWH in care.

## Conclusions

Our study identified significant gaps in measurement of HBV DNA and suppression of HBV viremia among PWH with HBV coinfection in the US and Canada. Periodic evaluation of the HBV care cascade among persons with HIV/HBV will be critical to monitoring success in completion of each step.

## Supporting information

**S1 Checklist. STROBE statement—Checklist of items that should be included in reports of observational studies.**
(DOCX)

## Acknowledgments

We are grateful for the participants and representatives of the NA-ACCORD Collaborating Cohorts:

**AIDS Clinical Trials Group Longitudinal Linked Randomized Trials:** Constance A. Benson and Ronald J. Bosch

**AIDS Link to the IntraVenous Experience:** Gregory D. Kirk

**Emory-Grady HIV Clinical Cohort:** Vincent Marconi and Jonathan Colasanti

**Fenway Health HIV Cohort:** Kenneth H. Mayer and Chris Grasso

**HAART Observational Medical Evaluation and Research:** Robert S. Hogg, Viviane Lima, P. Richard Harrigan, Julio SG Montaner, Benita Yip, Julia Zhu, Kate Salters, and Karyn Gabler

**HIV Outpatient Study:** Kate Buchacz and Jun Li

**HIV Research Network:** Kelly A. Gebo and Richard D. Moore

**Johns Hopkins HIV Clinical Cohort:** Richard D. Moore

**John T. Carey Special Immunology Unit Patient Care and Research Database, Case Western Reserve University:** Jeffrey Jacobson

**Kaiser Permanente Mid-Atlantic States**: Michael A. Horberg

**Kaiser Permanente Northern California:** Michael J. Silverberg

**Longitudinal Study of Ocular Complications of AIDS:** Jennifer E. Thorne

**MACS/WIHS Combined Cohort Study:** Todd Brown, Phyllis Tien, and Gypsyamber D'Souza

**Maple Leaf Medical Clinic:** Graham Smith, Mona Loutfy, and Meenakshi Gupta

**The McGill University Health Centre, Chronic Viral Illness Service Cohort:** Marina B. Klein

Multicenter Hemophilia Cohort Study–II: Charles Rabkin

**Ontario HIV Treatment Network Cohort Study:** Abigail Kroch, Ann Burchell, Adrian Betts, and Joanne Lindsay

**Parkland/UT Southwestern Cohort:** Ank Nijhawan

**Retrovirus Research Center, Universidad Central del Caribe, Bayamon Puerto Rico:** Angel M. Mayor

**Southern Alberta Clinic Cohort:** M. John Gill

**Study of the Consequences of the Protease Inhibitor Era:** Jeffrey N. Martin

**Study to Understand the Natural History of HIV/AIDS in the Era of Effective Therapy:** Jun Li and John T. Brooks

**University of Alabama at Birmingham 1917 Clinic Cohort:** Michael S. Saag, Michael J. Mugavero, and James Willig

**University of California at San Diego:** Laura Bamford, Edward Cachay, and Maile Karris

**University of North Carolina at Chapel Hill HIV Clinic Cohort:** Joseph J. Eron and Sonia Napravnik

**University of Washington HIV Cohort:** Mari M. Kitahata and Heidi M. Crane

**Vanderbilt Comprehensive Care Clinic HIV Cohort:** Timothy R. Sterling, David Haas, Peter Rebeiro, Austin Katona, and Megan Turner

**Veterans Aging Cohort Study:** Lesley Park and Amy Justice

NA-ACCORD Study Administration:

**Executive Committee:** Richard D. Moore, Keri N. Althoff, Stephen J. Gange, Mari M. Kitahata, Jennifer S. Lee, Michael S. Saag, Michael A. Horberg, Marina B. Klein, Rosemary G. McKaig, and Aimee M. Freeman

**Administrative Core:** Richard D. Moore, Keri N. Althoff, and Aimee M. Freeman

**Data Management Core:** Mari M. Kitahata, Stephen E. Van Rompaey, Heidi M. Crane, Liz Morton, Justin McReynolds, and William B. Lober

**Epidemiology and Biostatistics Core:** Stephen J. Gange, Jennifer S. Lee, Brenna Hogan, Bin You, Elizabeth Humes, Lucas Gerace, Cameron Stewart, and Sally Coburn

**Disclaimer:** The content is solely the responsibility of the authors and does not necessarily represent the views of the National Institutes of Health.

## Author Contributions

**Conceptualization:** Jessica Kim, H. Nina Kim, Vincent Lo Re, III.

**Formal analysis:** Craig W. Newcomb.

**Funding acquisition:** Vincent Lo Re, III.

**Methodology:** Jessica Kim, Craig W. Newcomb, Dean M. Carbonari, Jessie Torgersen, H. Nina Kim, Vincent Lo Re, III.

**Project administration:** Dean M. Carbonari.

**Supervision:** Vincent Lo Re, III.

**Visualization:** Jessica Kim, Craig W. Newcomb, Dean M. Carbonari, H. Nina Kim, Vincent Lo Re, III.

**Writing – original draft:** Jessica Kim, Craig W. Newcomb, Dean M. Carbonari, Jessie Torgersen, H. Nina Kim, Vincent Lo Re, III.

**Writing – review & editing:** Jessica Kim, Craig W. Newcomb, Dean M. Carbonari, Jessie Torgersen, Keri N. Althoff, Mari M. Kitahata, Marina B. Klein, Richard D. Moore, K. Rajender Reddy, Michael J. Silverberg, Angel M. Mayor, Michael A. Horberg, Edward R. Cachay, Joseph K. Lim, M. John Gill, Kara Chew, Timothy R. Sterling, Mark Hull, Eric C. Seaberg, Gregory D. Kirk, Sally B. Coburn, Raynell Lang, Kathleen A. McGinnis, Kelly A. Gebo, Sonia Napravnik, H. Nina Kim, Vincent Lo Re, III.

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
