## [Decision Letter · Decision Letter 0]

6 Jun 2023

PONE-D-23-11889Hepatitis B care cascade among people with HIV/HBV coinfection in the North American AIDS Cohort Collaboration on Research and Design, 2012-2016PLOS ONE

Dear Dr. Lo Re,

Thank you for submitting your manuscript to PLOS ONE. After careful consideration, we feel that it has merit but does not fully meet PLOS ONE’s publication criteria as it currently stands. Therefore, we invite you to submit a revised version of the manuscript that addresses the points raised by both reviewers during the review process. Please submit your revised manuscript by Jul 21 2023 11:59PM. If you will need more time than this to complete your revisions, please reply to this message or contact the journal office at plosone@plos.org. Please include the following items when submitting your revised manuscript:A rebuttal letter that responds to each point raised by the academic editor and reviewer(s). You should upload this letter as a separate file labeled 'Response to Reviewers'.A marked-up copy of your manuscript that highlights changes made to the original version. You should upload this as a separate file labeled 'Revised Manuscript with Track Changes'.An unmarked version of your revised paper without tracked changes. You should upload this as a separate file labeled 'Manuscript'.

We look forward to receiving your revised manuscript.

Kind regards,

Wenyu Lin, PhD

Academic Editor

PLOS ONE

“J.T. reports grants to her institution from the City of Philadelphia ID/SUD Care Integration Pilot. K.N.A. reports grants to her institution from the National Institute of Health (NIH), royalties from Coursera, and consulting fees from NIH and TrioHealth. M.B.K. reports grants from Canadian Institutes of Health Research, Fonds de recherché Quebec –Sante, NIH, ViiV Health Care, Gilead Sciences, and Abbvie; consulting fees from ViiV Health Care, Gilead Sciences, and Abbvie; leadership role in the CIHR Canadian HIV Trials Network; and receipt of goods/services from Siga Technologies. K.R.R. reports grants to his institution from Mallinckrodt, Exact Sciences, BMS, Intercept, Merck, Gilead, Grifols, Sequana, HCC-TARGET, NASH-TARGET, and BioVie; royalties from UpToDate; consulting fees from Spark Therapeutics, Mallinckrodt, Genfit, and Novo Nordisk; paid board participation from Novartis; and leadership roles in Gastroenterology and AASLD Task Force for COVID Activities. E.R.C. reports grants to his institution from Gilead Sciences and board participation in THERAtechnologies. J.K.L. reports grants to his institution from Intercept, Gilead, Viking, Pfizer, Eiger, Inventiva, and Novo Nordisk and leadership roles in the American Association for Study of Liver Diseases, American Gastroenterological Association, and American College of Gastroenterology. M.J.G. reports participation on the HIV national advisory boards for Merck, Gilead, and Viiv. K.C. reports grants to her institution from Merck Sharp & Dohme and Amgen; consulting fees from Pardes Bioscences; honoraria payments from International Antiviral Society-USA; and participation in the UCSF Safety Monitoring Committee. M.H. reports grants from Gilead Life Science for an investigator initiated study and participation in the data safety monitoring board for the M2HepPreP study. G.D.K. reports grants to his institution from NIH. K.A.G. reports grants to her institution from NIH, US Department of Defense, Defense Health Agency, State of Maryland, Octapharma, Mental Wellness Foundation, HealthNetwork Foundation, Bloomberg Philanthropies, and Moriah Fund; royalties from UpToDate; consulting fees from Spark HealthCare, Teach for America, and Aspen Institute; and unpaid advisor participation on a Pfizer scientific advisory board. H.N.K. reports grants to her institution from Gilead Sciences. V.L.R. reports grants to his institution from NIH and a leadership role in the International Society for Pharmacoepidemiology. All other authors reported no conflicts of interest.”

Reviewers' comments:

Reviewer's Responses to Questions

**Comments to the Author**

1. Is the manuscript technically sound, and do the data support the conclusions?

Reviewer #1: No

Reviewer #2: Partly

2. Has the statistical analysis been performed appropriately and rigorously? 

Reviewer #1: No

Reviewer #2: N/A

3. Have the authors made all data underlying the findings in their manuscript fully available?

Reviewer #1: Yes

Reviewer #2: Yes

4. Is the manuscript presented in an intelligible fashion and written in standard English?

Reviewer #1: Yes

Reviewer #2: Yes

5. Review Comments to the Author

Reviewer #1: In this article, authors only listed the data.A depth analysis and mining of these data is needed to conduct , otherwise the results obtained will be meaningless.So, the author's conclusion is not credible. Because of lacking meaningful statistical analysis in the article.

Reviewer #2: The authors have addressed a clinically relevant question; namely, the importance to monitor HBV DNA levels during therapy. There are, however, more details are needed to draw the conclusions.

Major comments:

1) It is uncertain if the patients had HIV-related and other lab tests more frequently than the HBV DNA measurements during the duration of observation. It is, therefore, unclear whether the providers did not order the HBV DNA tests or the patients did not go for the tests.

2) For those who had HBV DNA testing, how frequently were they tested?

3) It is important to document that these patients were compliant in taking their HBV therapy.

4) Besides monitoring for HBV DNA suppression, HCC surveillance is also essential in the hepatitis B care cascade.

6. PLOS authors have the option to publish the peer review history of their article (what does this mean?). If published, this will include your full peer review and any attached files.

Reviewer #1: No

Reviewer #2: No

---

## [Author Response · Author response to Decision Letter 0]

17 Jul 2023

Editorial Comments

Response: We have ensured that our manuscript meets PLOS ONE’s style requirements, including file naming.

2. Thank you for completing the Competing Interests section. Please confirm that this does not alter your adherence to all PLOS ONE policies on sharing data and materials, by including the following statement: "This does not alter our adherence to PLOS ONE policies on sharing data and materials.”

Response: We give PLOS ONE permission to add the requested sentence to the end of the Competing Interests section: “The stated competing interests do not alter our adherence to PLOS ONE policies on sharing data and materials.”

Response: The North American AIDS Cohort Collaboration on Research and Design (NA-ACCORD) maintains restrictions on sharing de-identified data sets. The NA‐ACCORD Principals of Collaboration requires submission and approval of a concept sheet that describes the intended research project for which data are being requested. The NA‐ACCORD Executive Committee and the Steering Committee (composed of principle investigators from each of the contributing cohorts) must approve the concept sheet and elect to have their data included in the research project. A signed Data User Agreement is required before data can be released. Guidance for how to obtain NA‐ACCORD data are outlined on the NA‐ACCORD website (https://naaccord.org/collaboration-policies).

Reviewer 1

1. In this article, the authors only listed the data. A depth analysis and mining of these data is needed, otherwise the results obtained will be meaningless. The authors’ conclusions are not credible because the manuscript is lacking a meaningful statistical analysis in the article.

Response: In May 2016, the World Health Assembly adopted the Global Health Sector Strategy on Viral Hepatitis, which called for the elimination of viral hepatitis as a public health threat by 2030 [1]. Elimination was defined as a 90% reduction in incidence and a 65% reduction in the number of related deaths from levels as of 2015 [2]. The strategy addressed all five hepatitis viruses (i.e., hepatitis A, B, C, D, and E), but hepatitis B virus (HBV) was of particular focus because of its public health burden. 

Monitoring progress towards these global targets across different settings will be crucial to the elimination of HBV infection [3, 4]. Care cascades have emerged as a critical tool for evaluating the delivery of care across sequential stages of management of chronic viral infections, starting with diagnosis of the infection and ending with viral suppression or cure [5-10]. The HIV care cascade (consisting of steps for diagnosis, linkage to care, retention in care, prescription of antiretroviral therapy [ART], and viral suppression) has been an effective tool for improving the health of people with HIV (PWH) and for achieving the public health benefits of ART [5-7]. The hepatitis C virus (HCV) care cascade (consisting of steps for diagnosis, confirmatory HCV RNA testing, prescription of direct-acting antiviral therapy, and viral cure) has been used to assess the delivery of HCV-related care in a variety of settings and has been important for monitoring progress toward HCV elimination goals [8-10]. Despite the descriptive nature of cascade of care data, the results are meaningful for establishing baseline metrics and progress in care. They can also identify key gaps in the delivery of care and enable national, regional, and local agencies to prioritize and target valuable resources to increase engagement along the cascade. Notably, further mining of data is not necessary to create a cascade of care.

In contrast, there have been few data describing the HBV care cascade, particularly among PWH with HBV coinfection [11]. This information is important to identify gaps in HBV care among PWH, target allocation of health resources to close those gaps, and promote stakeholder involvement/collaboration, which supports achievement of the 2030 World Health Organization HBV elimination goals [12]. We now clarify this important background in the revised Introduction on pages 4-5 (lines 70-96). We have also now added a detailed Statistical Analysis section on pages 7-8 (lines 151-171) to describe the analytic approach and enhance the transparency and reproducibility of our results.

Reviewer 2

1. It is uncertain if the patients had HIV-related lab tests more frequently than the HBV DNA measurements during the duration of observation. It is, therefore, unclear whether the providers did not order the HBV DNA tests or the patients did not go for the tests.

Response: To assess whether patients had HIV RNA measured more frequently than HBV DNA, we now report the median (interquartile range [IQR]) number of HIV RNA measures per patient between 2012-2016 on page 11 (lines 217-219). The median number of HBV DNA measures per patient between 2012-2016 for those who had this test performed was 3 (IQR, 2-6). In contrast, the median number of HIV RNA measures per patient between 2012-2016 was 8 (IQR, 4-12).

2. For those who had HBV DNA testing, how frequently were they tested? 

Response: On page 11 (lines 217-218), we now report the median (IQR) number of HBV DNA measures per patient between 2012-2016 for those who had this test performed. The median number of HBV DNA measures per patient between 2012-2016 for those who had this test performed was 3 (IQR, 2-6).

3. It is important to document that these patients were compliant in taking their HBV therapy.

Response: We agree that it is important to determine if patients are adherent to their HBV therapy. Measures of adherence to antiviral therapies, such as ART or HCV therapy, that are assessed within observational studies (e.g., self-report or pharmacy-based refill measures) can be inaccurate and are typically validated against the gold standard of viral suppression [13-16]. Consequently, as the final step of our HBV care cascade, we assessed the proportion of PWH with HBV coinfection who had an undetectable HBV DNA test during HBV treatment. On pages 13-14 (lines 270-279) of the Discussion, we now acknowledge that the assessment of adherence to HBV-active ART was beyond the scope of the present work and acknowledge this as a limitation. We note that future studies should determine the levels of adherence to HBV-active ART or entecavir required to achieve HBV DNA suppression in PWH, which could serve as a target for patients to maximize their response to HBV therapy.

4. Besides monitoring for HBV DNA suppression, hepatocellular carcinoma (HCC) surveillance is also essential in the hepatitis B care cascade.

Response: We did not include HCC surveillance as a step in our HBV care cascade because not all PWH with HBV coinfection are currently recommended to undergo HCC surveillance. According to guidelines by the American Association for the Study of Liver Diseases [17], the European Association for the Study of the Liver and European Organization for Research and Treatment of Cancer [18], and the Asian-Pacific Association for the Study of the Liver [19], HCC surveillance is only recommended for patients with chronic HBV infection and cirrhosis as well as those without cirrhosis who have specific characteristics, such as family history of HCC or certain age thresholds, sex, or race.

If HCC surveillance were a universal indication among PWH with chronic HBV, it would have made sense to include this activity as a step in the cascade. However, since HCC surveillance is not currently recommended for all PWH with HBV coinfection, we did not include it as a step in the cascade. We now clarify this decision in the revised Discussion on page 13 (lines 260-269).

REFERENCES

1. World Health Organization. Global Health Sector Strategy on Viral Hepatitis 2016-2021: Towards Ending Viral Hepatitis. Accessed at: https://apps.who.int/iris/handle/10665/246177. Accessed on: July 7, 2023.

2. World Health Organization. Global Hepatitis Report, 2017. Accessed at: https://www.who.int/publications/i/item/9789241565455. Accessed on: July 10, 2023.

3. Cui F, Blach S, Manzengo Mingiedi C, Gonzalez MA, Sabry Alaama A, Mozalevskis A, et al. Global reporting of progress towards elimination of hepatitis B and hepatitis C. Lancet Gastroenterol Hepatol. 2023;8(4):332-42. Epub 2023/02/11. doi: 10.1016/S2468-1253(22)00386-7. PubMed PMID: 36764320.

4. McMahon BJ. Sliding down the cascade of care for chronic hepatitis B virus infection. Clin Infect Dis. 2016;63(9):1209-11. Epub 2016/08/04. doi: 10.1093/cid/ciw517. PubMed PMID: 27486113.

5. Gardner EM, McLees MP, Steiner JF, Del Rio C, Burman WJ. The spectrum of engagement in HIV care and its relevance to test-and-treat strategies for prevention of HIV infection. Clinical infectious diseases : an official publication of the Infectious Diseases Society of America. 2011;52(6):793-800. Epub 2011/03/04. doi: 10.1093/cid/ciq243. PubMed PMID: 21367734; PubMed Central PMCID: PMC3106261.

6. Vital signs: HIV prevention through care and treatment--United States. MMWR Morbidity and mortality weekly report. 2011;60(47):1618-23. Epub 2011/12/02. PubMed PMID: 22129997.

7. Gardner EM, McLees MP, Steiner JF, Del Rio C, Burman WJ. The spectrum of engagement in HIV care and its relevance to test-and-treat strategies for prevention of HIV infection. Clin Infect Dis. 2011;52(6):793-800. Epub 2011/03/04. doi: 10.1093/cid/ciq243. PubMed PMID: 21367734; PubMed Central PMCID: PMCPMC3106261.

8. Yehia BR, Schranz AJ, Umscheid CA, Lo Re V, 3rd. The treatment cascade for chronic hepatitis C virus infection in the United States: a systematic review and meta-analysis. PLoS One. 2014;9(7):e101554. doi: 10.1371/journal.pone.0101554. PubMed PMID: 24988388; PubMed Central PMCID: PMCPMC4079454.

9. Safreed-Harmon K, Blach S, Aleman S, Bollerup S, Cooke G, Dalgard O, et al. The consensus hepatitis C cascade of care: Standardized reporting to monitor progress toward elimination. Clin Infect Dis. 2019;69(12):2218-27. Epub 2019/07/29. doi: 10.1093/cid/ciz714. PubMed PMID: 31352481.

10. Ferrante ND, Newcomb CW, Forde KA, Leonard CE, Torgersen J, Linas BP, et al. The hepatitis C care cascade Dduring the direct-acting antiviral era in a United States commercially insured population. Open Forum Infect Dis. 2022;9(9):ofac445. Epub 2022/09/13. doi: 10.1093/ofid/ofac445. PubMed PMID: 36092829; PubMed Central PMCID: PMCPMC9454032.

11. Jain MK, Opio CK, Osuagwu CC, Pillai R, Keiser P, Lee WM. Do HIV care providers appropriately manage hepatitis B in coinfected patients treated with antiretroviral therapy? Clin Infect Dis. 2007;44(7):996-1000. Epub 2007/03/08. doi: 10.1086/512367. PubMed PMID: 17342656.

12. World Health Organization. Combating hepatitis B and C to reach elimination by 2030. Accessed at: https://apps.who.int/iris/bitstream/handle/10665/206453/WHO_HIV_2016.04_eng.pdf;jsessionid=12C9049334C7AF699445190FCEF64A12?sequence=1. Accessed on March 1, 2023.

13. Miller LG, Hays RD. Measuring adherence to antiretroviral medications in clinical trials. HIV Clin Trials. 2000;1(1):36-46. Epub 2001/10/09. doi: 10.1310/enxw-95pb-5ngw-1f40. PubMed PMID: 11590488.

14. Grossberg R, Zhang Y, Gross R. A time-to-prescription-refill measure of antiretroviral adherence predicted changes in viral load in HIV. J Clin Epidemiol. 2004;57(10):1107-10. Epub 2004/11/06. doi: 10.1016/j.jclinepi.2004.04.002. PubMed PMID: 15528063.

15. Lo Re V, 3rd, Amorosa VK, Localio AR, O'Flynn R, Teal V, Dorey-Stein Z, et al. Adherence to hepatitis C virus therapy and early virologic outcomes. Clin Infect Dis. 2009;48(2):186-93. Epub 2008/12/18. doi: 10.1086/595685. PubMed PMID: 19086908; PubMed Central PMCID: PMCPMC2668718.

16. Lo Re V, 3rd, Teal V, Localio AR, Amorosa VK, Kaplan DE, Gross R. Relationship between adherence to hepatitis C virus therapy and virologic outcomes: a cohort study. Ann Intern Med. 2011;155(6):353-60. Epub 2011/09/21. doi: 10.7326/0003-4819-155-6-201109200-00003. PubMed PMID: 21930852; PubMed Central PMCID: PMCPMC3366635.

17. Marrero JA, Kulik LM, Sirlin CB, Zhu AX, Finn RS, Abecassis MM, et al. Diagnosis, staging, and management of hepatocellular carcinoma: 2018 practice guidance by the American Association for the Study of Liver Diseases. Hepatology. 2018;68(2):723-50. Epub 2018/04/07. doi: 10.1002/hep.29913. PubMed PMID: 29624699.

18. European Association for the Study of the Liver. Electronic address eee, European Association for the Study of the L. EASL Clinical Practice Guidelines: Management of hepatocellular carcinoma. J Hepatol. 2018;69(1):182-236. Epub 2018/04/10. doi: 10.1016/j.jhep.2018.03.019. PubMed PMID: 29628281.

19. Omata M, Cheng AL, Kokudo N, Kudo M, Lee JM, Jia J, et al. Asia-Pacific clinical practice guidelines on the management of hepatocellular carcinoma: a 2017 update. Hepatol Int. 2017;11(4):317-70. Epub 2017/06/18. doi: 10.1007/s12072-017-9799-9. PubMed PMID: 28620797; PubMed Central PMCID: PMCPMC5491694.

---

## [Decision Letter · Decision Letter 1]

16 Aug 2023

Hepatitis B care cascade among people with HIV/HBV coinfection in the North American AIDS Cohort Collaboration on Research and Design, 2012-2016

PONE-D-23-11889R1

Dear Dr. Lo Re,

We’re pleased to inform you that your manuscript has been judged scientifically suitable for publication and will be formally accepted for publication once it meets all outstanding technical requirements.

Kind regards,

Wenyu Lin, PhD

Academic Editor

PLOS ONE

Additional Editor Comments (optional):

The authors have adequately addressed reviewer's comments. The manuscript is suitable for publication in PLOS One.

Reviewers' comments:

Reviewer's Responses to Questions

**Comments to the Author**

1. If the authors have adequately addressed your comments raised in a previous round of review and you feel that this manuscript is now acceptable for publication, you may indicate that here to bypass the “Comments to the Author” section, enter your conflict of interest statement in the “Confidential to Editor” section, and submit your "Accept" recommendation.

Reviewer #1: All comments have been addressed

2. Is the manuscript technically sound, and do the data support the conclusions?

Reviewer #1: Yes

3. Has the statistical analysis been performed appropriately and rigorously? 

Reviewer #1: Yes

4. Have the authors made all data underlying the findings in their manuscript fully available?

Reviewer #1: Yes

5. Is the manuscript presented in an intelligible fashion and written in standard English?

Reviewer #1: Yes

6. Review Comments to the Author

Reviewer #1: Unlike hepatitis C virus care cascade, hepatitis B virus care cascade is a very complicated problem. That is because, If the hepatitis C virus cannot be detected, it can be considered that hepatitis C virus has been eliminated. However, the fact that undetectable quantitative hepatitis B virus does not mean that HBV has been eliminated, it may only be suppressed. Many patients with undetected hepatitis B virus will become HBV DNA positive while drug use is stopped,

Therefore, it is unreasonable to only use the undetected HBV DNA as the end point of monitoring (step4) in hepatitis B virus care cascade. But this irrationality is not caused by author design, but because there is no drug to eradicate hepatitis B virus in the current scientific and technological development. So I suggest this article is acceptable.

7. PLOS authors have the option to publish the peer review history of their article (what does this mean?). If published, this will include your full peer review and any attached files.

Reviewer #1: No

---

## [Editor Report · Acceptance letter]

24 Aug 2023

PONE-D-23-11889R1 

Hepatitis B care cascade among people with HIV/HBV coinfection in the North American AIDS Cohort Collaboration on Research and Design, 2012-2016 

Dear Dr. Lo Re:

I'm pleased to inform you that your manuscript has been deemed suitable for publication in PLOS ONE. Congratulations! Your manuscript is now with our production department. 

Kind regards, 

on behalf of

Dr. Wenyu Lin 

Academic Editor

PLOS ONE